# Relationship between Insulin Resistance (HOMA-IR), Trabecular Bone Score (TBS), and Three-Dimensional Dual-Energy X-ray Absorptiometry (3D-DXA) in Non-Diabetic Postmenopausal Women

**DOI:** 10.3390/jcm9061732

**Published:** 2020-06-03

**Authors:** Francisco Campillo-Sánchez, Ricardo Usategui-Martín, Ángela Ruiz -de Temiño, Judith Gil, Marta Ruiz-Mambrilla, Jose María Fernández-Gómez, Antonio Dueñas-Laita, José Luis Pérez-Castrillón

**Affiliations:** 1Gynaecology Department, Hospital Clínico Universitario Valladolid, 47003 Valladolid, Spain; fdacampillo@gmail.com; 2IOBA, University of Valladolid, 47011 Valladolid, Spain; 3Department Medicine, University of Valladolid, 47005 Valladolid, Spain; angelatemi@hotmail.com (Á.R.-d.T.); mruizm65@gmail.com (M.R.-M.); dueton@gmail.com (A.D.-L.); 4Hospital Nuestra Señora de Sonsoles, 05004 Avila, Spain; jgdayla@hotmail.com; 5Biology Department, University of Valladolid, 47005 Valladolid, Spain; josefg@med.uva.es; 6Department of Medicine, University of Valladolid Service of Clinical Toxicology, Río Hortega University Hospital, 47012 Valladolid, Spain; 7Department of Internal Medicine, Department of Medicine, University of Valladolid, Río Hortega University Hospital, 47012 Valladolid, Spain

**Keywords:** HOMA-IR, HbA1c, vBMD, TBS, 3D-DXA

## Abstract

Background: Insulin may play a key role in bone metabolism, where the anabolic effect predominates. This study aims to analyze the relationship between insulin resistance and bone quality using the trabecular bone score (TBS) and three-dimensional dual-energy X-ray absorptiometry (3D-DXA) in non-diabetic postmenopausal women by determining cortical and trabecular compartments. Methods: A cross-sectional study was conducted in non-diabetic postmenopausal women with suspected or diagnosed osteoporosis. The inclusion criteria were no menstruation for more than 12 months and low bone mass or osteoporosis as defined by DXA. Glucose was calculated using a Hitachi 917 auto-analyzer. Insulin was determined using an enzyme-linked immunosorbent assay (EIA). Insulin resistance was estimated using a homeostasis model assessment of insulin resistance (HOMA-IR). DXA, 3D-DXA, and TBS were thus collected. Moreover, we examined bone parameters according to quartile of insulin, hemoglobin A1C (HbA1c), and HOMA-IR. Results: In this study, we included 381 postmenopausal women. Women located in quartile 4 (Q4) of HOMA-IR had higher values of volumetric bone mineral density (vBMD) but not TBS. The increase was higher in the trabecular compartment (16.4%) than in the cortical compartment (6.4%). Similar results were obtained for insulin. Analysis of the quartiles by HbA1c showed no differences in densitometry values, however women in Q4 had lower levels of TBS. After adjusting for BMI, statistical significance was maintained for TBS, insulin, HOMA-IR, and HbA1c. Conclusions: In non-diabetic postmenopausal women there was a direct relationship between insulin resistance and vBMD, whose effect is directly related to greater weight. TBS had an inverse relationship with HbA1c, insulin, and insulin resistance unrelated to weight. This might be explained by the formation of advanced glycosylation products (AGEs) in the bone matrix, which reduces bone deformation capacity and resistance, as well as increases fragility.

## 1. Introduction

Insulin binds to their receptors in pre-osteoblasts and osteoblasts by stimulating two metabolic pathways: mitogen-activated protein kinase (MAPK) and phosphatidylinositol 3-kinase pathway (PI3-K/AKT), which favor the growth, proliferation, and survival of these cells [1,2]. Insulin behaves like an anabolic hormone that increases bone mass. Osteoblast activation produces the release of receptor activator of NF-kB ligand (RANKL) that binds to osteoclast receptor activator of NF-kB (RANK) receptors located. This simulates their proliferation and maturation. Thus, insulin plays a key role in bone metabolism [3].

Hyperinsulinism and insulin resistance are the key etiopathogenic elements of type 2 diabetes. Hyperinsulinism could be responsible for the increase in bone mass observed in diabetic patients [4] and non-diabetic postmenopausal women [5]. However, not all studies show uniform results. Some report a positive association between circulating insulin levels and bone mineral density (BMD), which did not change after weight adjustment [6]. Others show an association that disappears when weight is adjusted [7,8], while others show differing results, i.e., a lack of association or negative association between insulin and bone mass [9]. Despite this increase in cortical and trabecular bone mass, patients with type 2 diabetes have an increased fracture risk [10]. Factors related to bone quality are likely to be involved [10].

There have been recently developed procedures that analyze bone quality, such as the trabecular bone score (TBS). TBS is derived from the lumbar dual-energy X-ray absorptiometry (DXA) images and uses software to analyze grayscale. It correlates with three-dimensional parameters of bone microarchitecture such as trabecular connectivity, the number of trabeculae, and their separation [11]. A decline in the TBS is associated with an increased risk of fracture [12]. Three-dimensional (3D) densitometry permits an assessment of the shape and intrinsic material properties that determine bone strength. Hip DXA is transformed using segmentation algorithms to provide a 3D analysis of the cortical and trabecular compartments [13].

The aim of this study was to analyze the relationship between insulin resistance and bone quality measured using TBS and three-dimensional dual-energy X-ray absorptiometry (3D-DXA) in non-diabetic postmenopausal women by determining the cortical and trabecular compartments, as well as how this is linked to body mass index (BMI) and age.

## 2. Patients and Methods

### 2.1. Patients

A cross-sectional study was conducted in non-diabetic postmenopausal women who were suspected or diagnosed with osteoporosis. The inclusion criteria were postmenopausal women (no menstruation for more than 12 months) and a low bone mass or osteoporosis, as defined by DXA. We randomly included individuals from Rio Hortega University Hospital’s Densitometry Unit (Spain). Patients had been diagnosed with osteoporosis or suspected of low bone mass by clinical criteria according to the National Osteoporosis Foundation’s Clinician’s Guide to Prevention and Treatment of Osteoporosis [14]. A DXA was appointed for patients with this diagnosis. The exclusion criteria were type 1 and 2 diabetes, and lack of informed consent. Diagnosis of diabetes was established according to the American Diabetes Association (ADA) 2019 criteria (hemoglobin A1C (HbA1C) ≥ 6.5% and/or glucose ≥ 126 mg/dl). A protocol collected demographic data, lifestyle factors, previous illnesses, and past and present medication. The BMI was calculated. A venous sample was taken by venipuncture between 8 a.m. and 11 a.m. after 8 h of fasting. The serum was immediately frozen at −20 °C prior to testing. Glucose was calculated using a Hitachi 917 auto-analyzer. Insulin was determined using an enzyme-linked immunosorbent assay (EIA). Insulin resistance was estimated using the homeostasis model assessment of insulin resistance (HOMA-IR or HOMA model: fasting glucose × fasting insulin).

### 2.2. Three-Dimensional Dual-Energy X-ray Absorptiometry (3D-DXA)

A DXA scan was performed using a Prodigy scanner (GE healthcare, Madison, WI, USA), according to the manufacturer’s recommendations. The software 3D-SHAPER (2.6 version, Galgo Medical S.L, Barcelona, Spain) was also used. This method used a statistical 3D model of the proximal femur’s form and density, and was built from the quantitative computed tomography (QCT) database (Caucasian men and women). The details of this method’s modeling can be found in Winzenrieth et al. and Humbert et al. [12,13]. The study was made from DXA exploration in order to obtain a 3D model specific to the patient’s proximal femur, generating measurements in 3D from the total area of interest in the femur. The volumetric BMD (vBMD, mg/cm^3^), bone mineral content (BMC) (g), and volume (cm^3^) were calculated in the trabecular, cortical, and integral (trabecular + cortical) compartments, respectively. The cortical thickness (Cth, mm) and BMD of the cortical surface (sBMD cortical, mg/cm^2^, obtained by the multiplication of cortical vBMD (mg/cm^3^) and Cth (cm)) provided additional analysis for the cortical region. The models and 3D-SHAPER measurements’ precision were evaluated against a QCT [12,13]. The average form precision—i.e., the average distance between external limits of the femur geometry—were derived from 3D-SHAPER and QCT, and the result was 0.93 mm. Regarding bone density and cortical bone thickness, the correlation coefficients between 3D-SHAPER and the measurements derived from QCT were 0.86, 0.93, and 0.91 for trabecular vBMD, cortical vBMD, and cortical thickness, respectively [12,13].

### 2.3. Trabecular Bone Score

TBS was evaluated at the lumbar level (L1-L4) using TBSiNsight 2.1 (Med-Imaps, Merignac, France).

### 2.4. Statistical Analysis

Continuous variables were expressed as the mean ± standard deviation (SD), while categorical variables were expressed as the absolute (n) and relative (%) frequencies. A chi-squared test was used to compare the categorical variables. The distribution of variables was analyzed using the Kolmogorov–Smirnov test. In the case of parametric variables, we applied an analysis of variance (ANOVA) t-test. In the case of non-parametric variables, groups were compared using the Mann–Whitney U test (two groups) or Kruskal–Wallis test (more than two groups). Insulin, HOMA-IR, glucose, and HbA1c were stratified in quartiles. *P*-values were adjusted according to past and present medication, lifestyle factors, and the presence of osteoporosis and osteoporosis risk factors. A multiple linear regression was applied to study the association between different clinical parameters (i.e., BMI, insulin, HOMA-IR, glucose, and HbA1c) and the densitometry parameters. In the multiple linear regression, we established three models that depend on the adjusted parameters. In Model 1, the linear regression analyses were adjusted by past and present medication, lifestyle factors, and the presence of osteoporosis and osteoporosis risk factors. In Model 2, they were adjusted by past and present medication, lifestyle factors, age, and the presence of osteoporosis and osteoporosis risk factors. Lastly, in Model 3, they were adjusted by past and present medication, lifestyle factors, age, BMI, and the presence of osteoporosis and osteoporosis risk factors. We applied a Bonferroni adjustment in all comparisons. A value of *p* < 0.05 was considered significant. All analyses were performed using the SPSS version 22.0 statistical package (SPSS, Chicago, IL, USA).

### 2.5. Ethical Aspects

The experimental protocol was approved by the Río Hortega University Hospital of Valladolid Ethics Committee and complied fully with the Declaration of Helsinki (2008) of the World Medical Association and Spanish data protection law (LO 15/1999) and specifications (RD 1720/2007). All patients who agreed to participate gave signed written consent.

## 3. Results

Our study sample included 381 non-diabetic postmenopausal women (mean age 62 ± 9 years, mean BMI 26 ± 4). The mean age of menopause was 47 ± 6 years and the menarche was 13 ± 1.5 years. There were 29% (n = 115) smokers and 1.8% (n = 7) with excessive alcohol intake. There were 6.82% (n = 26) in treatment with thiazides, 11.02% (n = 42) with serotonin receptor inhibitors, 0.26% (n = 1) with androgenic inhibitors, 3.67% (n = 14) with beta blockers, 9.7% (n = 37) with thyroid hormone, 19.42% (n = 74) with bone antiresorptives, 1.04% (n = 4) with TSH, 2.88% (n = 11) with anabolic therapy, 10.23% (n = 39) with corticosteroids, and 2.62% (n = 10) were treated with strontium ranelate. Mean glucose values were 90 ± 24 mg/dl, mean insulin 13 ± 14 UI/l, mean HOMA-IR 3.3 ± 4.6, and mean glycated hemoglobin A1c (HbA1c) 5.6 ± 0.4%.

Table 1 summarizes the clinical characteristics between osteoporotic and non-osteoporotic patients. In the preliminary analysis, we separately assessed osteoporotic individuals. There were no differences between osteoporotic and non-osteoporotic patients except that osteoporotic women were older. In addition, all the osteoporotic patients were treated and we did not find significant statistical differences in the osteoporotic group in the comparison, according the quartiles of HOMA-IR, insulin, and HbA1c (Table 2). Therefore, all participants were analyzed globally. The results were adjusted by the presence of osteoporosis.

Table 2 shows the general patient characteristics, dividing the patient’s HOMA-IR, insulin, and HbA1C by quartiles. The women in the fourth quartile (Q4) of HOMA-IR and insulin had a higher BMI, and there were differences in smoking in the two groups. When we considered HbA1C, in addition to BMI, they had an older age. The results of the analysis of densitometry parameters divided patients according to quartiles of HOMA-IR, insulin, and HbA1c (Table 3). Women in Q4 HOMA-IR had higher values of vBMD but not TBS. The increase was higher in the trabecular compartment (16.4%) than in the cortical compartment (6.4%). The data for insulin were similar (Table 3). When the quartiles were assessed by HbA1c, no significant differences were observed in densitometry values. Yet in the TBS, women in Q4 had lower levels (Table 3). Table 4 summarizes the results of the linear regression analysis of the relationship between BMI, insulin, HOMA-IR, glucose, HBA1c, and the densitometry parameters. We established three models that depend on the adjusted parameters. In Model 1, the linear regression analyses were adjusted by past and present medication, lifestyle factors, and the presence of osteoporosis and osteoporosis risk factors. In Model 2, they were adjusted by past and present medication, lifestyle factors, age, and the presence of osteoporosis and osteoporosis risk factors. Lastly, in Model 3, they were adjusted by past and present medication, lifestyle factors, age, BMI, and the presence of osteoporosis and osteoporosis risk factors. While assessing the general characteristics of our population, we obtained a significant difference in BMI in our samples (Table 2). This fact, in combination with the literature on the importance of weight in bone quality and structure, lead us to include it and adjust it accordingly.

Comparing the results in our models, there were barely any differences between Model 1 and 2 in statistical significance. However, in Model 3, statistical significance disappeared in the densitometry parameters (BMD total, Sdensitometry, vBMD trabecular, vBMD cortical, vBMD integral, and mCT). Our results showed that when adjusting for BMI, the statistical significance was maintained for insulin, HOMA-IR, TBS, and HbA1c. BMI had a direct and significant relationship with glucose, insulin, HOMA-IR, and HbA1c (Table 4).

## 4. Discussion

We found that higher values of HOMA-IR and insulin were associated with increases in vBMD, at both the cortical and trabecular levels. However, after adjusting for BMI, this relationship disappeared. The vBMD can be used to determine bone material properties like stiffness and strength, standing for a great asset of bone fracture prediction [15]. QCT is the gold standard for assessing volumetric density in the cortical and trabecular compartments [16,17]. However, QCT, compared to DXA, exposes the patient to higher radiation. Device availability is limited and the cost is relatively high, which limits the use of QCT for routine patient explorations and monitoring [18]. Recently, 3D models were generated from DXA images [19,20]. These new techniques—i.e., 3D modeling—have been proposed to solve these limitations [12]. These techniques use a statistical 3D shape and a density model of the proximal femur constructed from a database of QCT scans. The model is subsequently recorded from a standard hip DXA. Humbert et al. [19] used DXA projections to extrapolate 3D density distributions for trabecular and cortical regions, as well as the femoral shape of the femur and cortical thickness. For the extrapolated vBMD in the trabecular and cortical regions and for the mean cortical thickness, they found correlation coefficients of 0.86, 0.93, and 0.91 between 3D-DXA and QCT measurements, respectively. Biomechanical descriptors obtained through FE simulations integrate femur shape, cortical thickness, and volumetric distribution of BMD using a 3D-Shaper. These improve the discrimination of facture occurrence, especially when compared to the classical use of areal BMD or vBMD. Indeed, one compartment of the two (trabecular or cortical) can be more impacted than the other [21] and the two compartments can react differently [22,23]. The potential of this new approach can be a good surrogate of QCT, in the context of osteoporosis diagnosis and drug treatment monitoring. The fact that DXA is less invasive than QTC makes a DXA-based 3D finite element (FE) model possible, especially in clinical practice as a routine patient screening.

Other studies have analyzed this relationship, but none has used the technique employed in the present study. To our knowledge, we are the first to report the relationship between 3D-DXA parameters according to insulin, HbA1c, and HOMA-IR in non-diabetic postmenopausal women. Shanbhogue et al. found an association between insulin resistance and total, trabecular, and cortical vBMD in White American women using HR-QCT [6]. Their effect was not modified after adjusting for weight. Haffner et al. found a direct relationship between insulin levels and vBMD in the femoral neck, yet the relationship disappeared after weight adjustment [24]. Similar results were obtained by Srikanthem et al. [7] and Napoli et al. [8], yet these study included 19% of diabetic patients. Other authors observed an inverse relationship with lower femoral neck strength relative load, while Asian studies found different results [9,25]. Our results showed that after adjusting for BMI, the negative association between vBMD and insulin resistance disappeared, which indicates that obesity is a key element. A small increase was observed at the cortical level and it is important to adjust the results by taking BMI into consideration, as we do in Model 3. None of our patients were diabetic although there was a high percentage of osteoporotic patients. We analyzed osteoporotic and non-osteoporotic patients together and our results showed no differences between them.

It has been suggested that being overweight, expressed through BMI, has protective effects on the skeleton [26], which might be explained by various mechanisms [27]. Leptin, a cytokine produced by adipocytes, is higher in individuals with major fat content. In vitro studies have shown that leptin stimulates osteoblastogenesis without affecting mature osteoblasts [28]. However, excess weight could be harmful due to the release of inflammatory cytokines by the visceral adipose tissue [29]. Interleucin-6 (IL-6) and tumor necrosis factor-alpha (TNF-∝) increase the expression of c-fmc gene, RANK, and RANKL, all of which stimulate osteoclastogenesis. The osteocyte behaves like a mechanostat in response to the mechanical overload of being overweight. It releases IGF-I, which acts on receptors located in the osteoblasts, thus increasing bone formation [30]. In turn, this molecule blocks the action of sclerostin, an inhibitor of the metabolic Wnt pathway. A direct relationship between sclerostin and insulin resistance has been found in obese patients and patients with type 2 diabetes [31,32]. Sclerostin inhibits bone formation but the effect on bones may be limited by the release of IGF-I. Our results showed a relationship between TBS, HOMA-IR, and HbA1c that was maintained after being adjusted by other parameters, including BMI. Women in the upper quartile of HbA1c had the lowest TBS values. As TBS measures bone quality and predicts the risk of fracture [33,34], numerous studies in diabetics have shown a decline in TBS that is inversely related to insulin resistance. In our study of a non-diabetic population, we found that women in Q4 of HbA1c had the lowest TBS levels, which was maintained after adjusting for age and BMI. Recently, there have been two studies published that analyzed the relationship between HOMA-IR- and TBS-measured insulin resistance in non-diabetic populations [35,36]; their results were similar to ours. Although their sample populations were both heterogeneous (including both men and women) and pre- and postmenopausal, they also had a small sample.

HbA1c assesses the metabolic situation in relation to blood glucose. Elevated glycaemia facilitates the non-enzymatic glycation that forms intramolecular bonds at the level of collagen, which forms the bone matrix, leading to the formation of so-called advanced glycosylation products (AGEs) [37]. AGEs are located in the middle region of collagen fibers and reduce bone deformation capacity as well as decrease resistance, which thus increases fragility. AGEs act on osteoblasts and osteoclasts. Moreover, they act on specific osteoblast receptors and decrease their proliferation and differentiation. They also activate the NF-kb pathway in osteoblasts, increasing the release of inflammatory cytokines that act on osteoclasts, thus stimulating their proliferation and activation, and increasing bone resorption [38]. These findings have some congruence with studies of TBS in type 1 diabetes where a relationship between fracture and TBS and fracture and HbA1c have been described [39].

Although our study provides a comprehensive evaluation of this topic, there were some limitations, which were determined by the absence of osteoporotic women without treatment. Moreover, we did not have data regarding the analysis of phosphocalcic metabolism or bone remodeling markers. The strengths were determined by its sample size, the homogeneity of the studied population, and to the best of our knowledge we are the first group to report on the relationship between 3D-DXA parameters according to insulin, HbA1c, and HOMA-IR in non-diabetic postmenopausal women.

In conclusion, we found a direct relationship between insulin resistance and vBMD in non-diabetic postmenopausal patients, whose effect is directly related to higher weight and had the greatest effect at the trabecular level. Thus, bodyweight is fundamental when evalutating a patient’s overall fracture risk. TBS was inversely related to HbA1c and unrelated to weight, indicating a deterioration in bone quality that could justify an increase in fracture rusk due to glycation affecting the bone architecture and impairing its correct healing.

## Figures and Tables

**Table 1 jcm-09-01732-t001:** General characteristics of the non-osteoporotic and osteoporotic subjects.

	Non-Osteoporotic	Osteoporotic	*p*-Value
Age, mean ± SD (years)	5987 ± 8.19	63.41 ± 8.89	<0.001
BMI, mean ± SD (Kg/m^2)^	2618 ± 4.56	25.64 ± 3.83	0.226
Age of menopause, mean ± SD (years)	4810 ± 5.68	47.23 ± 5.85	0.163
Smoking, n (%)	62 (53.9%)	53 (46.1%)	0.106
Alcohol, n (%)	4 (57.1%)	3 (42.9%)	0.457
Corticosteroids, n (%)	18 (46.2%)	21 (53.8%)	0.663
Familial history of osteoporosis, n (%)	53 (40.5%)	78 (59.5%)	0.427
Familial history of hip fracture, n (%)	20 (34.5%)	38 (65.5%)	0.147
Previous falls, n (%)	7 (35.0%)	13 (65.0%)	0.434

BMI: Body mass index.

**Table 2 jcm-09-01732-t002:** General characteristics of the study population stratified by quartiles of HOMA-IR, insulin and HbA1c.

GENERAL CHARACTERISTICS	HOMA-IR				
	Q1 (0.2–1)	Q2 (1.1–1.85)	Q3 (1.86–3.4)	Q4 (3.5–41.8)	*p*-Value
Age, mean ± SD (years)	61.84 ± 8.78	60.53 ± 8.57	61.87 ± 8.61	62.04 ± 8.62	0.657
BMI, mean ± SD (Kg/m^2^)	23.79 ± 3.09	25.12 ± 3.01	26.18 ± 3.81	27.62 ± 4.80	<0.001
Age of menopause, mean ± SD (years)	47.62 ± 5.96	47.63 ± 6.41	47.69 ± 5.73	47.34 ± 6.27	0.986
Osteoporosis, n (%)	36 (21.4%)	44 (26.2%)	48 (28.6%)	40 (23.8%)	0.560
Smoking, n (%)	22 (34.9%)	35 (36.5%)	20 (20.8%)	19 (19.8%)	0.037
Alcohol, n (%)	2 (33.3%)	1 (16.7%)	2 (33.3%)	1 (16.7%)	0.797
	INSULIN				
	Q1 (1.3–5.1)	Q2 (5.2–8.6)	Q3 (8.7–16.1)	Q4 (16.2–131.3)	*p*-value
Age, mean ± SD (years)	62.32 ± 9.25	60.67 ± 8.84	61.18 ± 8.63	61.67 ± 8.37	0.661
BMI, mean ± SD (Kg/m^2^)	24.14 ± 3.07	25.38 ± 3.51	26.31 ± 3.97	27.11 ± 4.74	<0.001
Age of menopause, mean ± SD (years)	47.94 ± 5.96	48.07 ± 5.90	46.77 ± 6.28	47.38 ± 6.05	0.575
Osteoporosis, n (%)	43 (25.1%)	47 (27.5%)	39 (22.8%)	42 (24.6%)	0.982
Smoking, n (%)	24 (24.2%)	38 (38.4%)	19 (19.2%)	18 (18.2%)	0.024
Alcohol, n (%)	2 (33.3%)	3 (50.0%)	1 (16.7%)	0 (0%)	0.414
	HbA1c				
	Q1 (4.04–5.3)	Q2 (5.5–5.5)	Q3 (5.6–5.7)	Q4 (5.8–8)	*p*-value
Age, mean ± SD (years)	58.61 ± 9.61	68.52 ± 8.07	62.61 ± 8.41	63.65 ± 8.15	<0.001
BMI, mean ± SD (Kg/m^2^)	24.90 ± 3.64	24.56 ± 2.95	25.98 ± 3.24	27.39 ± 4.87	<0.001
Age of menopause, mean ± SD (years)	47.33 ± 6.29	48.74 ± 4.79	47.45 ± 6.49	47.20 ± 6.16	0.457
Osteoporosis, n (%)	27 (18.1%)	33 (22.1%)	43 (28.9%)	46 (30.9%)	0.730
Smoking, n (%)	19 (21.8%)	26 (29.9%)	17 (19.5%)	25 (28.7%)	0.218
Alcohol, n (%)	1 (16.7%)	1 (16.7%)	3 (33.3%)	2 (33.3%)	0.959

BMI: Body mass index. HbA1c: Glycated hemoglobin A1c. HOMA-IR: homeostasis model assessment insulin resistance.

**Table 3 jcm-09-01732-t003:** Densitometry parameters according quartiles of HOMA-IR, insulin, and HbA1c.

DENSITOMETRY PARAMETERS	HOMA-IR				
	Q1 (0.2–1)	Q2 (1.1–1.85)	Q3 (1.86–3.4)	Q4 (3.5–41.8)	*p*-Value
Neck BMD (g/cm^2^)	0.805 ± 0.13	0.827 ± 0.11	0.840 ± 0.13	0.858 ± 0.17	0.316
Hip BMD (g/cm^2^)	0.822 ± 0.13	0.865 ± 0.12	0.876 ± 0.12	0.914 ± 0.14	0.001 Q1 vs. Q4
Sdensitometry (mg/cm^2^)	139 ± 22	147 ± 20	150 ± 21	155 ± 23	0.025, Q1 vs. Q3 0.001 Q1 vs. Q4
vBMD trabecular (g/cm^3^)	134 ± 38	144 ± 33	146 ± 35	156 ± 39	0.005 Q1 vs. Q4
vBMD cortical (g/cm^3^)	763 ± 82	792± 75	791 ± 74	812 ± 84	0.003 Q1 vs. Q4
vBMD integral(g/cm^3^)	279 ± 53	294 ± 47	297 ± 48	310 ± 52	0.003 Q1 vs. Q4
mCT (mm)	1.82 ± 0.13	1.86 ± 0.13	1.90 ± 0.16	1.90 ± 0.13	0.007 Q1 vs. Q3 0.003 Q1 vs. Q4
TBS	1.269 ± 0.16	1.267 ± 0.11	1.266 ± 0.14	1.269 ± 0.13	0.864
	INSULIN				
	Q1 (1.3–5.1)	Q2 (5.2–8.6)	Q3 (8.7–16.1)	Q4 (16.2–131.3)	*p*-value
Neck BMD (g/cm^2^)	0.800 ± 0.12	0.842 ± 0.11	0.841 ± 0.14	0.853 ± 0.17	0.125
Hip BMD (g/cm^2^)	0.824 ± 0.13	0.878 ± 0.13	0.879 ± 0.13	0.902 ± 0.13	0.001 Q1 vs. Q4
Sdensitometry (mg/cm^2^)	140 ± 21	148± 21	151 ± 22	153 ± 22	0.017 Q1 vs. Q3 0.002 Q1 vs. Q4
vBMD trabecular (g/cm^3^)	136 ± 36	146 ± 34	146 ± 38	153 ± 31	0.039 Q1 vs. Q4
vBMD cortical (g/cm^3^)	767 ± 80	790 ± 79	796 ± 76	806 ± 82	0.021 Q1 vs. Q4
vBMD integral(g/cm^3^)	281 ± 51	296 ± 49	298 ± 53	307 ± 48	0.014 Q1 vs. Q4
mCT (mm)	1.82 ± 0.13	1.87 ± 0.14	1.89 ± 0.16	1.89 ± 0.13	0.011 Q1 vs. Q3 0.008 Q1 vs. Q4
TBS	1.266 ± 0.16	1.275 ± 0.10	1.268 ± 0.10	1.249 ± 0.14	0.651
	HbA1c				
	Q1 (4.04–5.3)	Q2 (5.5–5.5)	Q3 (5.6–5.7)	Q4 (5.8–8)	*p*-value
Neck BMD (g/cm^2^)	0.836 ± 0.13	0.839 ± 0.13	0.853 ± 0.17	0.830 ± 0.11	0.926
Hip BMD (g/cm^2^)	0.859 ± 0.14	0.868 ± 0.12	0.876 ± 0.13	0.898 ± 0.13	0.304
Sdensitometry (mg/cm^2^)	145 ± 22	147 ± 22	148 ± 22	153 ± 22	0.160
vBMD trabecular (g/cm^3^)	145 ± 36	146 ± 35	144 ± 37	153 ± 31	0.446
vBMD cortical (g/cm^3^)	776 ± 74	783 ± 68	794 ± 87	811 ± 83	0.073
vBMD integral(g/cm^3^)	293 ± 50	295 ± 47	295 ± 52	306 ± 52	0.374
mCT (mm)	1.86 ± 0.13	1.88 ± 0.17	1.87 ± 0.11	1.88 ± 0.13	0.676
TBS	1.298 ± 0.11	1.280 ± 0.14	1.296 ± 0.10	1.229 ± 0.12	0.013,Q1 vs. Q4

HOMA-IR: homeostasis model assessment-insulin resistance. HbA1c: glycated hemoglobin A1c. BMD: bone mineral density. vBMD: volumetric bone mineral density. mCT: micro computed tomography. TBS: trabecular bone score.

**Table 4 jcm-09-01732-t004:** Lineal regression analysis between the different parameters (BMI, insulin, HOMA-IR, glucose, and HBA1c) and the densitometry parameters. In Model 1, the linear regression analyses were adjusted by past and present medication, lifestyle factors, and the presence of osteoporosis and osteoporosis risk factors. In Model 2, they were adjusted by past and present medication, lifestyle factors, age, and the presence of osteoporosis and osteoporosis risk factors. Lastly, in Model 3, they were adjusted by past and present medication, lifestyle factors, age, BMI, and the presence of osteoporosis and osteoporosis risk factors.

		Model 1		Model 2		Model 3	
		ρ	*p*-Value	ρ	*p*-Value	ρ	*p*-Value
BMI	BMD neck (g/cm^2^)	0.009	0.053	0.195	<0.001	-	-
	BMD total (g/cm^2^)	0.087	<0.001	0.187	<0.001	-	-
	Sdensitometry (mg/cm^2^)	0.128	<0.001	0.193	<0.001	-	-
	vBMD trabecular (g/cm^3^)	0.067	<0.001	0.181	<0.001	-	-
	vBMD cortical (g/cm^3^)	0.106	<0.001	0.157	<0.001	-	-
	vBMD integral (g/cm^3^)	0.078	<0.001	0.181	<0.001	-	-
	mCT (mm)	0.075	<0.001	0.12	<0.001	-	-
	TBS	0.024	0.003	0.083	<0.001	-	-
Insulin	BMD neck (g/cm^2^)	−0.002	0.541	0.162	0.289	0.221	0.738
	BMD total (g/cm^2^)	0.013	0.03	0.085	0.012	0.228	0.573
	Sdensitometry (mg/cm^2^)	0.015	0.021	0.052	0.012	0.223	0.565
	vBMD trabecular (g/cm^3^)	0.009	0.057	0.084	0.026	0.209	0.75
	vBMD cortical (g/cm^3^)	0.002	0.102	0.037	0.068	0.176	0.933
	vBMD integral (g/cm^3^)	0.009	0.054	0.082	0.024	0.212	0.674
	mCT (mm)	0.015	0.021	0.036	0.013	0.137	0.365
	TBS	0.086	0.005	0.021	0.008	0.091	0.009
HOMA-IR	BMD neck (g/cm^2^)	−0.003	0.673	0.165	0.413	0.224	0.633
	BMD total (g/cm^2^)	0.016	0.018	0.088	0.007	0.225	0.382
	Sdensitometry (mg/cm^2^)	0.016	0.017	0.053	0.01	0.217	0.466
	vBMD trabecular(g/cm^3^)	0.012	0.033	0.091	0.014	0.21	0.472
	vBMD cortical (g/cm^3^)	0.008	0.064	0.039	0.043	0.17	0.689
	vBMD integral (g/cm^3^)	0.011	0.037	0.085	0.017	0.21	0.494
	mCT (mm)	0.013	0.028	0.034	0.019	0.132	0.426
	TBS	0.016	0.019	0.094	0.011	0.1	0.022
Glucose	BMD neck (g/cm^2^)	0	0.322	0.149	0.345	0.195	0.998
	BMD total (g/cm^2^)	0.026	0.002	0.09	0.002	0.203	0.065
	Sdensitometry (mg/cm^2^)	0.025	0.003	0.055	0.003	0.206	0.112
	vBMD trabecular (g/cm^3^)	0.024	0.003	0.095	0.003	0.189	0.068
	vBMD cortical (g/cm^3^)	0.023	0.004	0.049	0.004	0.162	0.088
	vBMD integral (g/cm^3^)	0.023	0.004	0.089	0.004	0.194	0.089
	mCT (mm)	0.01	0.045	0.027	0.047	0.13	0.487
	TBS	−0.003	0.849	0.074	0.699	0.081	0.854
HbA1c	BMD neck (g/cm^2^)	−0.002	0.53	0.136	0.099	0.18	0.705
	BMD total (g/cm^2^)	0.028	0.004	0.091	<0.001	0.218	0.061
	Sdensitometry (mg/cm^2^)	0.025	0.007	0.056	0.011	0.222	0.189
	vBMD trabecular (g/cm^3^)	0.021	0.011	0.092	0.001	0.2	0.094
	vBMD cortical (g/cm^3^)	0.03	<0.001	0.057	0.001	0.185	0.071
	vBMD integral (g/cm^3^)	0.02	0.014	0.083	0.002	0.2	0.127
	mCT (mm)	0.004	0.167	0.02	0.085	0.129	0.909
	TBS	0.031	0.003	0.091	0.01	0.096	0.032

**ρ**: Correlation coefficient.

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
