# Peer review of "Relationship between Insulin Resistance (HOMA-IR), Trabecular Bone Score (TBS), and Three-Dimensional Dual-Energy X-ray Absorptiometry (3D-DXA) in Non-Diabetic Postmenopausal Women"

_jcm, 2020, doi:10.3390/jcm9061732_

Round 1
Reviewer 1 Report
The authors have gone an overall good job in revising the manuscript based on the reviewers comments; however, there are still modifications needed.
The major comments are below:
- The term "suspicion of osteoporosis" is used often and only defined once late in the methods. It appears to be related to low bone mass. This phrase is not clinically sound, and should change to "women with low bone mass and osteoporosis as defined by DXA."
- The write up of the results is very limited. With three different models, it would be nice for the authors to walk the reader through how estimates changed between the crude and the various adjusted models.
- The discussion section says "there is an inverse relationship between HbA1c and TBS," this implies as HbA1c increases, TBS decreases; however this is not what is shown in the tables. Please revise to match the results in the table.
Minor comments:
- 3D was used in the introduction before it was defined
- QCT is misspelled in the methods
- The statistical analysis section refers to quantitative and qualitative results; i believe you mean continuous and categorical variables based on the text.
Author Response
May 15th, 2020
Dear Sir,
We thank the Editor and Reviewer (s) for this opportunity to revise our manuscript based on their comments. We believe that the revisions based on their recommendations have improved the value and accuracy of our manuscript. The point-by-point answers to reviewers’ comments are provided below. We hope that with the modifications included in the manuscript it will be suitable for publication in your journal.
Response to Reviewer 1:
The major comments are below:
The term "suspicion of osteoporosis" is used often and only defined once late in the methods. It appears to be related to low bone mass. This phrase is not clinically sound, and should change to "women with low bone mass and osteoporosis as defined by DXA."
We corrected the term in the inclusion criteria as instructed.
The write up of the results is very limited. With three different models, it would be nice for the authors to walk the reader through how estimates changed between the crude and the various adjusted models.
We explained our models and how they affect the results more in depth in the Results section.
The discussion section says "there is an inverse relationship between HbA1c and TBS," this implies as HbA1c increases, TBS decreases; however, this is not what is shown in the tables. Please revise to match the results in the table.
Our results showed that women in Q4 of HbA1c had the lowest levels of TBS (Table 3). Following the suggestion of the reviewer, we have clarified this idea in the results and discussion sections.
Minor comments:
3D was used in the introduction before it was defined
We have changed it in text.
QCT is misspelled in the methods
We have corrected it.
The statistical analysis section refers to quantitative and qualitative results; i believe you mean continuous and categorical variables based on the text.
We have changed how we refer to the variables in the text.
Yours sincerely,
Prof. José Luis Pérez-Castrillón MD PhD on behalf of the authors
Department of Internal Medicine, Rio Hortega University Hospital.
Department of Medicine, University of Valladolid.
Valladolid, Spain

Reviewer 2 Report
the Authors have improved the manuscript significantly, however there are still some issues that need to be addressed.
- the English should still be improved. For instance what do you mean not having menstruation for 12 years after evolution
- in the result section only the BMI model is explained. Better explanation of the models and choosing the best one is recommended
- i am not convinced that pooling the osteoporotic and non osteoporotic individuals in the analysis is a good idea, especially because their mean age is significantly different. Have you tried assessing them separately? Was the results the same?
- the tables are still vague. For instance in the modelling table, the models should be explained in the footnote. The tables explaining the quartile are not easy to understand and not explained in the text.
- the discussion is much easier to follow but considering the title more focus on the modelling is recommended
Author Response
Dear Sir,
We thank the Editor and Reviewer (s) for this opportunity to revise our manuscript based on their comments. We believe that the revisions based on their recommendations have improved the value and accuracy of our manuscript. The point-by-point answers to reviewers’ comments are provided below. We hope that with the modifications included in the manuscript it will be suitable for publication in your journal.
Response to Reviewer 2:
The English should still be improved. For instance, what do you mean not having menstruation for 12 years after evolution
The manuscript has been revised by a native English medical translator and we have made the proper adjustments.
The line in the text that said “no menstruation for more than 12 months” is referring to the consensus diagnosis of menopause, which is a year with no period (Menopause: diagnosis and management. NICE Guideline, No. 23 London: National Institute for Health and Care Excellence (UK); 2019 Dec.) We have clarified this idea in the Patients and Methods section.
In the result section only the BMI model is explained. Better explanation of the models and choosing the best one is recommended
We added a more detailed explanation of the models in the Results section.
I am not convinced that pooling the osteoporotic and non osteoporotic individuals in the analysis is a good idea, especially because their mean age is significantly different. Have you tried assessing them separately? Were the results the same?
First we assessed them separately and the results were quite similar. In addition, all the osteoporotic patients were following treatment and we did not find significant statistical differences in the osteoporotic group in the comparison according to the quartiles of HOMA-IR, insulin and HbA1c (Table 2). Also, all the results were adjusted by the presence of osteoporosis. Therefore, we finally decided to use our sample as the whole. We have clarified it in the text.
The tables are still vague. For instance, in the modelling table, the models should be explained in the footnote. The tables explaining the quartile are not easy to understand and not explained in the text.
We corrected the footnotes and addressed the tables in the text.
The discussion is much easier to follow but considering the title more focus on the modelling is recommended
Following the recommendation of the Reviewer, we have addressed the modelling more in depth in the discussion.
Yours sincerely,
Prof. José Luis Pérez-Castrillón MD PhD on behalf of the authors
Department of Internal Medicine, Rio Hortega University Hospital.
Department of Medicine, University of Valladolid.
Valladolid, Spain

Round 2
Reviewer 1 Report
The authors have done a great job modifying the papers based on reviewer comments to produce a clear and sound manuscript to provide new data on the relationship between metabolic markers and bone health.
Reviewer 2 Report
Most of the comments are well addressed
This manuscript is a resubmission of an earlier submission. The following is a list of the peer review reports and author responses from that submission.
Round 1
Reviewer 1 Report
The authors have done a nice job studying the effect of HOMA-IR, TBS and 3D DXA on bone mass and quality. The manuscript though should be revised according to following comments.
The manuscript should be edited by a native speaker. There are a lot of grammatical and spelling mistakes and sometimes its hard to understand the phrase. The text should be revised so that the flow of the text is kept. It feels like the authors jump from one topic to the other one and so the reader is lost. The opening of the article gives the reader the feeling that they are starting in the middle of the text and some parts are missing beforehand In the methods, they suggest that they have tested women suspected of osteoporosis and osteoporotic ones. What do you mean by suspected of osteoporosis, and how is this determined? What about those who were already taking medication, where they also included? These medication could have affected your test results, how did you check their affects? In abstract, result section, what do you mean by women in Q4? Q4 of HOMA, insulin or ? In the methods section, you have to define how the non-diabetics and based on which guidelines were defined. A better explanation of how the 3D-DXA modl was built is needed. Its not clear what/and from whom are the scans in the QCT database? and how the final model was developed. I think this is the part that differs the current study from similar ones. In the results, what do you mean by all the osteoporotic patients were treated? Those who were diagnosed during the study? What do you mean by all participants were analyzed globally? In the last paragraph of the results, the authors mention that they have developed some modelings. These models are not explained in the results, not really clear in the table and not discussed in the discussion There are many other factors that influence osteoporosis, for instance medication (osteoporotic medication), how did the authors removed their confounding effects The discussion is really hard to follow. Normally you should report your results, explain what was shown in literature and if different what do you think is the result. The last two paragraphs, before conclusion, for sure do not belong to the end of discussion. They also some pathways here but fail to link it with the rest of the manuscript In discussion it is not always clear if the authors are reporting their results or that from literature. Again most of the time its not obvious why the results of the current study are different from literature and what do they show? For instance I liked to see the explanation why vBMD is affected by insulin and not TBS after weight adjustments.
Reviewer 2 Report
To provide more evidence in the association between insulin and metabolic factors associated with diabetes and its relationship with bone quality, the authors conducted a cross-sectional study evaluating metabolic factors and bone parameters in 381 non-diabetic post-menopausal women from Spain. DXA was used to asses bone density, 3D DXA and TBS were used to assess bone quality. The metabolic factors included: insulin, HOMA-IR, glucose, and HbA1c. Crude models showed significant associations between higher levels of metabolic biomarkers with higher bone parameters, but after adjustment for age and BMD, insulin, HOMA-IR, and HbA1C had significant associations with TBS. The results are interesting, but there are some issues warranting clarification. See comments below:
Major Issues:
1) The inclusion criteria were women with "suspicion or diagnosis" of osteoporosis; however, the authors did not specify what constitutes "suspicion" of osteoporosis. Was this women with low bone mass/osteopenia or was this women with fractures? This needs to be specified.
2) The title of the paper is "relationship between insulin resistance, TBS in non-diabetic women"; given this, why was BMI included as a independent variable? More justification on why this was included is needed.
3) The information in Table 6 needs clarification. Upon first review, I assumed the table was presenting beta coefficients and p-values, so that we were to interpret results as "for ever 1 unit increase in BMI femoral neck BMD increased by 0.009 g/cm2." However, the column heading and footnotes says the pearson correlation coefficient is being presented, which means the that "BMI explains 0.009% of the variance in femoral neck BMD in these non-diabetic women." Typically r values should not be negative as indicated for femoral neck BMD in the crude models of insulin, HOMA-IR, and HbA1c. Once the presentation of these results is corrected, the interpretation of these results needs modification.
Minor comments:
Abstract:
- Correct spelling of DXA in methods section
- Need to add statistical analysis in methods section; particularly tp support first statement in the conclusions section.
- If space allows, it would be great to have a brief description of the study sample at the beginning of the results section.
- The second sentence in the results section reference quartile 4, but does not reference what biomarker this is referring to.
Introduction:
- Sentence "Despite this increase in cortical and trabecular bone mass...." needs references
- I would spell out "dimensional" on your first reference to 3D DXA
- Your objectives state that you want to determine how it is linked to BMI; given BMI is regularly used clinically, it makes sense to use this metric, but given that you have DXA, you could potentially expand to evaluate the role of body composition measurements from DXA in the association. This could add more to the field.
Methods:
- I would reorganize the entire statistical analysis section to match tables.
- Descriptive statistics by osteoporosis status using t-test for continuous variables and chi2 test for categorical variables.
- Evaluated descriptives and bone densitometry parametrors by quartiles of biomarkers using Anova for continuous variables and chi2 for categorical variables. Boneferroni adjustments for all comparisons.
- These are currently in separate tables, but can be combined into one or have one table with descriptives and densitometry for each biomarker in one table
- hierarchical multiple linear regression models: model 1 crude, model 2: adjusted for age; model 3: adjusted for age and BMI.
- Text mentions adjusting for lifestyle factors and medications, but this was not listed in the table. If so, it needs to be listed.
Results:
- The reorganization of the statistical analysis section will assist in better clarifying how the results are presented in the results section. For example, all of the descriptive information should be in Table 1.
- Need to better walk through the main results, particular what is presented in table 6, particularly the full model results
Discussion:
- I does not appear that the results match what is written in the text. For example, "We found no relationship between insulin with TBS after adjusting for BMI;" whereas in Table 6 shows a significant p-value for TBS in every model. Likewise, the text says "There was a negative association between vBMD and insulin resistance," but table 6 only showed a negative values with femoral neck BMD. All of this needs to get clarified.
- The authors did not list any limitations of their study.
